# Progress in Plant Genome Sequencing

Robert J. Henry [1,2]

1   Queensland Alliance for Agriculture and Food Innovation, University of Queensland,
    Brisbane, QLD 4072, Australia; robert.henry@uq.edu.au
2   ARC Centre of Excellence for Plant Success in Nature and Agriculture, University of Queensland,
    Brisbane, QLD 4072, Australia

**Abstract:** The genome sequence of any organism is key to understanding the biology and utility of that organism. Plants have diverse, complex and sometimes very large nuclear genomes, mitochondrial genomes and much smaller and more highly conserved chloroplast genomes. Plant genome sequences underpin our understanding of plant biology and serve as a key platform for the genetic selection and improvement of crop plants to achieve food security. The development of technology that can capture large volumes of sequence data at low costs and with high accuracy has driven the acceleration of plant genome sequencing advancements. More recently, the development of long read sequencing technology has been a key advance for supporting the accurate sequencing and assembly of chromosome-level plant genomes. This review explored the progress in the sequencing and assembly of plant genomes and the outcomes of plant genome sequencing to date. The outcomes support the conservation of biodiversity, adaptations to climate change and improvements in the sustainability of agriculture, which support food and nutritional security.

**Keywords:** DNA sequencing; plant genome; long read sequencing; chromosome assembly

## 1. Introduction

Advances in the analysis of DNA sequences have been a key driver of enhanced biological understanding and the application of biological knowledge [1]. DNA sequencing in the 20th century was largely based on Sanger sequencing, which limited both the quality (accuracy) and volume of data that could be generated relative to the next generation sequencing that we have today [2]. The introduction and rapid development of next generation sequencing has resulted in an acceleration in the development of plant genome sequencing, especially over the last decade [3]. This technology has evolved rapidly, resulting in continuous major changes to the strategies that are used to sequence and assemble genomes. For example, when only short-read sequences were available, physical mapping was a key strategy. Large fragments of the genomes were cloned in bacterial artificial chromosomes (BACs) [4]. The BACs were then sequenced and the genomes were assembled by covering the genetic maps with BAC tiles [5]. The availability of accurate long read sequencing has made these approaches largely redundant [6]. A review in 2018 [7] reported that 236 angiosperm genome sequences had been reported. Since then, many more genomes have been sequenced and the quality of the genome sequences has increased significantly. The NCBI database (https://www.ncbi.nlm.nih.gov/genome/browse#!/overview/flowering%20plants; accessed 6 June 2022) includes 831 flowering plant genomes, with 373 at the chromosome level. The de novo assembly of long read sequences allows very large contigs to be assembled, sometimes representing a complete plant chromosome [8].

## 2. Diversity of Plant Genomes

Plant genomes vary enormously in size, even within closely related groups of plants [9]. The nuclear genomes of flowering plants (angiosperms) vary more than 1000-fold, from

less than 100 kb to more than 100 Gb [10]. The genomes of gymnosperms are generally large and complex and represent an even greater challenge for genome sequencing [11]. The large (10 Gb) genome of *Ginkgo biloba* has recently been reported [12], which provides the first reference genome for gymnosperms. Genomes also vary greatly in terms of their content of repetitive sequences, the level of gene duplication, their ploidy and their heterozygosity, providing a range of challenges and degrees of difficulty within genome sequencing and assembly.

## 3. Applications of Plant Genome Sequencing

### 3.1. Model Genomes

The challenge of sequencing plant genomes using early technologies made it necessary to focus on sequencing model genomes that could be used to study related, but more complex, species. The first plant to have a sequenced genome was *Arabidopsis thaliana* [13], which was chosen because it is a small plant with a rapid generation time and a very small genome, thereby making it an ideal model plant for research use. The first crop plant with a sequenced genome [14] was rice (*Oryza sativa*), which was chosen because it is a major food crop plant with a relatively small genome. This became a model for cereal and grass genomes. Similarly, *Brachypodium distachyon* was sequenced [15] as a model grass genome, which is especially relevant for the wheat genome. Recent advances in genome sequencing technology have greatly reduced the need for models as it is now possible to sequence most species easily.

### 3.2. Crop Plant Genomes

The sequencing of the genomes of crop species has become a key enabling tool for plant improvement. Most major crops now have reference genome sequences [16] and as the technology becomes more powerful and the costs reduce, genomes are also being generated for many other minor crops. This usually involves the production of a reference genome sequence for a species and the re-sequencing of many individuals to define allelic variations within that species. Current efforts recognize that a single reference genome cannot always serve the needs of plant breeders, so pan-genomes that capture the variations in many diverse genomes within the gene pool are being produced as breeding platforms.

### 3.3. Sequencing Plant Biodiversity

Many diverse plant genomes have now been sequenced with an increasing coverage of the major groups, especially among flowering plants. The coverage of plant orders is high and the genomes from many plant families have now been reported; however, coverage at the genus level is still very low for most plant groups. Systematic efforts to obtain plant genome sequences may take a top-down approach to sequencing a member of each plant family, then each genus and, finally, each species would become available as resources. Ultimately, the re-sequencing of the diversity within each species is of value. A knowledge of the diversity within plant populations is a fundamental tool that can guide the effective conservation of the diversity within species.

### 3.4. Sequencing Rare and Threatened Species

Targeted efforts are now being made to sequence rare and threatened species of plants as a tool to aid conservation, both in situ [17] and ex situ [18]. This is more urgent among critically endangered species, for which a genome sequence may be all we can retain as the species are lost to extinction. Efforts to sequence biodiversity often focus on rare species as the highest priority.

The critically endangered wild crop relative *Macadamia jansenii* has been used to compare plant genome sequencing and assembly methods [19]. This has allowed for the comparison of sequencing platforms and bioinformatics tools for genome assembly using a common sample. The generation of a chromosome-level genome sequence for a plant involves the preparation of a DNA sample, the sequencing of that DNA, the assembly of

the sequence reads into contigs and, finally, the assembly of the sequence contigs into a chromosome-level assembly (Figure 1).

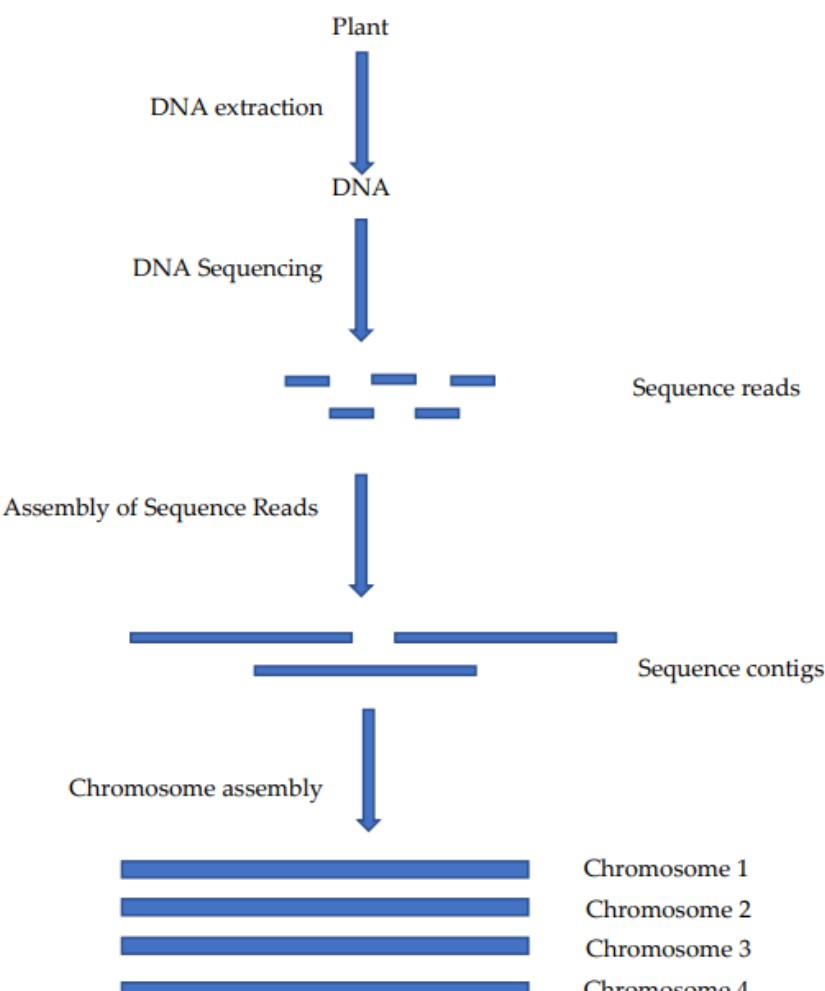

**Figure 1.** Steps in the sequencing and assembly of a plant genome: DNA extraction is used to produce a DNA sample that is suitable for sequencing, the sequencing of the DNA produces long read sequences, the reads are self-assembled into contigs (often at or near chromosome length) and these contigs are then assembled at the chromosome level using chromatin mapping or genetic mapping.

## 4. Sequencing Technology

### 4.1. DNA Isolation

The starting point for the sequencing of plant genomes is obtaining a sample of DNA to sequence. The ease of obtaining a DNA sample of suitable quality for sequencing varies greatly between species. Plants contain many secondary metabolites, proteins and polysaccharides that may interfere with DNA extraction and become a source of contaminants that often reduce the efficiency of DNA extraction. Current technologies require a minimum amount of DNA and the DNA must be pure (free from contaminants that may inhibit sequencing) to facilitate efficient sequencing and the generation of large volumes of data. The amount of DNA required for long read sequencing has been greater (usually μg quantities) than that needed for short read sequencing (https://dnatech.genomecenter.ucdavis.edu/pacbio-library-prep-sequencing; accessed on 6 June 2022). For long read sequencing, the DNA must be intact (not degraded) so that long sequences are present in the sample and can be sequenced to extract long reads.

Simple methods that were suitable for DNA extraction in the past for purposes such as PCR analysis [20] are no longer adequate, so the development of species- [21] or tissue-specific methods that can support next generation sequencing are often required [22]. Some species are especially difficult and require the isolation of the nucleus [23] first as a source of DNA that may be free from contaminants from other parts of the plant cell. The isolation of nuclei in plants is challenging for some of the same reasons that DNA isolation is difficult. The disruption of the plant cell wall requires forces that may damage organelles, such as nuclei and shear DNA.

*4.2. Sort Read Sequences*

The first set of next generation sequencing technologies provided lager volumes of short DNA sequences. The accuracy of these short sequences and the volume of data have since increased dramatically. The length of these sequences started at around 30 bp and has rapidly advanced to 100–150 bp. Paired-end sequencing has extended this technology to allow for the production of sequences of around 400 bp, but most applications currently deliver sequences of around 150 bp. Illumina sequencing platforms are the dominant technology used for short read sequencing. This technology conducts sequencing by synthesis in a very large number of parallel reactions. The incorporation of nucleotides is monitored as the DNA is copied. Other techniques (e.g., solid [24] and 454 sequencing [25]) have been replaced by new technologies because they generally offered lower accuracy or data volumes, resulting in relatively higher costs. Ion Torrent sequencing is used for the rapid determination of a sequence of large numbers of small sequences, such as amplicon sequencing and 16 S metagenomic sequencing. In plants, this has been used for chloroplast sequencing [26].

*4.3. Long Read Sequences*

The assembly of plant genomes with large numbers of repetitive sequences is not possible with only short read sequences. Therefore, technology that allows much longer sequences to be generated has been key to simplifying genome assembly. The length of these sequences and their accuracy have improved greatly since the technology was first introduced.

4.3.1. PacBio

Pacific Biosciences (PacBio) has developed a long-read sequencing platform that provides accurate long read sequencing. The single-molecule real-time (SMRT) sequencing involves monitoring the incorporation of fluorescent-labeled nucleotides [27]. Recently, single long reads (also known as continuous long reads (CLR)) have largely been replaced by HiFi reads, which provide a consensus sequence based on sequencing a long fragment of DNA (approximately 15,000 bp) multiple times by first circularizing the DNA and reading around the circle many times [28]. The repeated sequencing of the same molecule allows a highly accurate sequence to be generated as the circular consensus sequence (CCS) is read. The quality of the genomes that are generated by the assembly of these reads into contigs has been improved by the application of optimized assembly tools, such as those provided by hifiasm.

4.3.2. ONT

Oxford Nanopore Technologies (ONT) provides a long read sequencing technology that delivers accurate sequence data quickly. The sequence is determined by measuring the changes in electrical currents as the DNA is passed through a pore. The ONT platform generates very long reads and has the advantage of very low instrumentation costs. This platform has continued to improve and deliver very long read sequences with increasing accuracy [29]. ONT sequencing has been widely applied to very rapid sequencing, such as that required for diagnostics [30], due to the advantage of having portable instruments.

The chromosome-level assemblies of plant genomes can be achieved in combination with methods, such as optical mapping [31].

### 4.3.3. Other

Several technology providers have developed pseudo-long reads that are created by linking short reads. These techniques may produce long reads at lower costs, but the long reads that are generated often do not match the accuracy of the current long read methods [19]. These technologies have been developed by Universal Sequencing Technology [32], MGI [33] and 10× genomics [34]. Despite the great contribution that long read sequencing technology has made to the efficient production of high-quality plant genomes, the emergence of further advances in long read sequencing technologies remains one of the key areas that may contribute to future advances.

### 4.3.4. Advances

Genome sequencing and assembly requires an adequate depth of sequencing. The size of contigs that can be assembled as long read sequence data has been shown to increase in an almost linear way [8]. The size of the assembled genomes reduces slightly with more contiguous assemblies, probably due to the joining of homologous contig ends, as does the completeness [8]. Improved software has also enabled improvements in the assembly of long read sequences [35]. The use of hifiasm has been shown to allow the haplotype-resolved assembly of the large (30 Gb) genome of the Californian redwood (*Sequoia sempervirens*) [36]. These advances are illustrated by the quality of the early plant genomes relative to those that are being generated by the latest technology. The first rice genome, which was reported in 2002, was highly fragmented while current technology delivers sequence contigs that are often full-length chromosomes [35].

### 4.4. Chromosome-Level Assembly

The ultimate aim of genome sequencing is to obtain a complete genome sequence of each chromosome, from telomere to telomere. This relies on evidence from beyond the DNA sequence data. Physical and genetic mapping methods have been used to achieve the chromosome-level assembly of contigs that were generated from sequencing data [37]. Recently, the advances in sequencing technology have made it possible to generate many full-length chromosomes from the sequence data alone [35]. The complete assembly of sequence contigs into whole chromosomes has been widely achieved using genetic mapping data, chromatin mapping (Hi-C) or optical mapping. Hi-C [38] involves the mapping of chromatin by crosslinking the DNA in the intact chromatin, digesting the DNA and then sequencing (short reads) the DNA fragments at the ends of the crosslinks. These are used to position the sequence contigs along the chromosome. Optical mapping (Bionano) can also be used to locate sequences along the DNA sequence and to scaffold the sequence contigs [39]. Many projects have combined these technologies to support the generation of high-quality genomes. Recent advances in long read sequencing have enabled the generation of long contigs of highly accurate sequences, reducing reliance on these techniques for high-level assembly. They remain essential for the de novo assembly of most chromosome-level genomes. High-quality sequence contigs in combination with genetic mapping data, Bionano optical data or Hi-C chromatin mapping have generally succeeded in achieving chromosome-level assemblies of plant genomes. A report on more than 100 chromosome-level assemblies in 2021 [40] found only a 73% coverage of the pseudomolecules that represent the chromosomes. The combination of long read sequencing and the use of these tools has resulted in the recent reporting of many high-quality chromosome-level genome sequences (Table 1), with the quality improving greatly along with the most recent technology.

**Table 1.** Some recent chromosome-level assemblies of plant genomes.

| | Species | Technique * | Reference |
|---|---|---|---|
| Amorphophallus | *Amorphophallas konjac* | Hi-C | [41] |
| Apple | *Malus domestica* | Genetic Map | [42] |
| Avocado | *Persea americana* | Genetic Map | [43] |
| Banana | *Musa balbisiana* | Hi-C | [44] |
| Camphor | *Cinnamomum camphora* | Hi-C | [45] |
| Carrot | *Daucus carota* | Genetic Map | [46] |
| Chinese Skullcap | *Scutellaria baicalensis* | Hi-C | [47] |
| Crucihimalaya | *Crucihimalaya lasicocarpa* | Hi-C | [48] |
| Cucumber | *Cucumis metuliferus* | Hi-C | [49] |
| Eucalypt | *Corymbia citriodora* | Genetic Map | [37] |
| Field Pennycresss | *Thlaspi arvense* | Genetic Map/Hi-C/Bionano | [50] |
| Ginger | *Zingiber officinale* | Hi-C | [51] |
| Ginkgo | *Ginkgo biloba* | Hi-C | [52] |
| Jojoba | *Simmondsia chinensis* | Hi-C | [53] |
| Macadamia | *Macadamia jansenii* | Hi-C | [54] |
| | *Macadamia integrifolia* | Genetic Map | [55] |
| Paper Mulberry | *Broussonetis papyrifera* | Hi-C | [56] |
| Peach | *Prunus persica* | Hi-C | [57] |
| Peanut | *Arachis hypogaea* | Hi-C | [58] |
| Speranskia | *Speranskia yunnanensis* | Hi-C | [59] |
| Taxus | *Taxus chinensis* | Hi-C | [60] |
| Tea | *Camellia sinensus* | Hi-C | [61] |
| Water Caltrop | *Trapa* spp. | Hi-C | [62] |

* The technique used to achieve the chromosome-level assembly of the sequence contigs.

### 4.5. Haplotype-Resolved Genomes

Most published plant genomes are collapsed representations of the diploid genome as a single sequence, with a random inclusion of one of the two alleles at each heterozygous position. Only recently has it become possible to assemble each haplotype separately [63]. This has been the result of advances in both sequencing technology and sequence assembly tools. Current technology suggests that most genomes can now be sequenced at the haplotype level, thereby replacing the reporting of collapsed genomes with the sequences of the two haplotypes.

### 4.6. Pan-Genomes

The sequencing of plant genomes has shown that significant differences may be found within a plant species, which means that more than one reference genome is required to represent the species. The sequencing of plant genomes has also demonstrated that many genes are variably present in different individuals within a species. These presence/absence differences have led to the construction of pan-genomes, which represent the complete set of genes found within a population. A genome that includes all of the variations within a group of plants is known as a pan-genome. The pan-genome concept is a powerful tool for plant breeders for the analysis of gene pools [64]. Pan-genomes can be generated at different levels to represent the diversity that is found within, for example, domesticated gene pools, species or genera.

### 4.7. Transcriptomes

Transcriptome sequencing is an important tool for the analysis of the expressed regions of a genome. This is key to understanding gene functions and the determination of the genetic basis of important plant traits [65–70]. Transcriptome sequencing complements genome sequencing in genome characterization. Transcripts provide physical evidence that the sequence is formed of the expressed and complementary predictions of genes, based on the sequence alone. The comparison of the transcriptomes of different genotypes

from different tissues or cell types at different stages of development and under different environmental conditions allows for the discovery of the genes that control plant traits and has become a key approach in plant biology and the discovery of genes for selection in plant breeding. Single-cell transcriptomics has become a powerful tool for understanding gene expression at the cell and tissue level but has had limited application in plants [71], partly due to the difficulty in isolating specific plant cells without disrupting expression.

### 4.7.1. RNAseq

The quantitative analysis of the levels of expression of genes in any specific cell, tissue, organ, genotype or development stage is widely determined by RNA sequencing (RNAseq) [72,73]. RNAseq has largely replaced earlier array-based or gene by gene analysis tools as it provided a more unbiased analysis of the whole transcriptome.

An analysis of the gene expression in the highly polyploid sugarcane genome revealed that while the different alleles of most genes are expressed in direct proportion to their abundance in the genome, some genes show highly biased patterns of expression [74]. In hexaploid wheat, subgenome-specific responses to diseases have also been reported [75].

### 4.7.2. Long Read Transcriptomes

Long read sequencing is a method that has been applied to the analysis of plant transcriptomes, which reveals the diversity of full-length transcripts and defines the variations in splicing and intron retention in gene expression [76]. The long read sequencing of transcriptomes avoids the challenge of the assembly of many closely related transcripts from short reads. Unique 3′ and 5′ sequences may be separated by common intervening sequences, which creates the risk of incorrectly combining the ends of the transcripts when using short reads.

Some examples of the application of long read sequencing to the analysis of plant transcriptomes of increasing complexity can be found for polyploid species in Table 2.

**Table 2.** The long read sequencing of polyploid transcriptomes.

| Species | Ploidy | Reference |
| --- | --- | --- |
| Coffee | 4X | [76] |
| Wheat | 6X | [77] |
| Strawberry | 8X | [78] |
| Sugarcane | 12X | [79] |

### *4.8. Organelle Genome Sequencing*

Plant cells usually contain a single nucleus and many organelles, probably hundreds of mitochondria and thousands of chloroplasts. Sequencing the organelle genomes is complicated by the transfer of genes between these genomes. The nuclear genome often contains many insertions of large and small sequences of organellar genomes. Many early methods struggled to distinguish organellar gene sequences from those of copies that were inserted into the nuclear genome because they relied on PCR amplification or organelle separation [80]. Nuclear inserts may represent versions of organellar genomes that were transferred in the past and that have diverged since insertion.

### 4.8.1. Chloroplast Genomes

The chloroplast genomes of plants are highly conserved sequences of 100–150 Kb, containing around 100 genes [81]. The structure of most chloroplasts is similar, with four components including inverted repeats that separate large and small single-copy regions. Chloroplasts have been widely used in plant identification due to their presence in all green plants and the high copy numbers in the cell that simplify the detection of chloroplast sequences. Early approaches that relied on chloroplast isolation or PCR amplification were plagued by confusion due to the copies of chloroplast sequences in the nuclear

and mitochondrial genomes. Recent approaches [82] rely on the higher abundance of chloroplast genome sequence reads in short read sequence data to clearly distinguish the correct sequence of the relevant chloroplast [83]. The development of software tools now allows for the efficient extraction of accurate whole chloroplast genome sequences from even low (nuclear) coverage sequencing datasets. The annotation of chloroplast genomes that were generated in this way has resulted in the identification of around 100 genes with increasingly well-defined functions [84].

The sequencing of the maternal (e.g., chloroplast) and nuclear genomes of plants has frequently revealed discordant phylogenies [85–87], suggesting widespread reticulate evolution in plant populations (Table 3). Chloroplast transfers between species during rare events results in "chloroplast capture" by closely related species.

**Table 3.** Discordant phylogenies for chloroplast and nuclear genome sequences.

| Species | Reference |
|---|---|
| Apple Tribe | [88] |
| Eucalypts | [87] |
| Osmorhiza spp. | [89] |
| *Pedicularis* spp. | [90] |
| Sorghum | [85] |
| Rice (*Oryza*) | [86] |

### 4.8.2. Plant Mitochondrial Genomes

The mitochondrial genomes of plants [91] are much larger and less conserved than the chloroplast genomes and as a result, they have been much less studied than chloroplasts. The mitochondrial genome, as with the nuclear genome, may include sequences that were derived from the chloroplast that have been inserted into the genome at various times throughout its evolutionary history. Due to the relatively higher number of mitochondrial genomes in cells, these sequences are even more likely to be confused with chloroplast genome sequences than chloroplast sequences that were inserted into nuclear genomes.

## 5. Biological Understanding

Sequencing plant genomes provides an enhanced understanding of the biology of plant species. This knowledge informs the better conservation of biodiversity and sustainable use in agriculture and food production. Plant genomes may often explain the response of plants to the environment and may assist in improving the management of crops.

### 5.1. Whole Genome Duplications

Genome sequencing and assembly defines the presence of duplicated regions of the genome that are often the result of whole genome duplications during the evolutionary history of the species. In many species, evidence can be found for more than one duplication event. The analysis of these events can aid in the determination of evolutionary relationships. The selection of key genomes for sequencing allows evolutionary relationships to be defined. For example, the sequencing of the *Aristolochia fimbriata* genome revealed that this plant lacked the whole genome duplication that was reported in other magnoliid plants, placing it at a basal position in the angiosperm phylogenies [92].

### 5.2. Polyploid Challenges

The sequencing and assembly of a polyploid plant genome is complicated by the presence of many similar sequences that can be difficult to assemble. Sugarcane is an important global crop, but the high degree of ploidy (12X) in this species has resulted in it being one of the last major crop species to have a sequenced genome. Instead, a monoploid equivalent (based on BAC clones from sugarcane that cover the sorghum genome) has been widely used in sugarcane genomes due to the lack of a polyploid genome [5].

While a genome for the diploid Robusta coffee was reported in 2014 [93], the sequencing of the tetraploid Arabica coffee has been more difficult due to a genome that was based on a doubled diploid currently being used to characterize the origins of Arabica coffee (Arabica Genome Sequencing Consortium).

### 5.3. Genomics of Plants with Diverse Reproductive Biology

While most flowering plants are hermaphrodites with both male and female reproductive organs, separate male and female plants are found in dioecious species. There are dioecious plants in many plant families and they represent around 6% of all plant species. The differences between the genomes of male and female dioecious plants have not been fully characterized. The recent sequencing of the male and female genomes of jojoba revealed large differences between the sex chromosomes, with the presence of many sex-specific genes [53]. The chromosome-level assembly of male and female genomes has provided a perspective on the basis of sexual dimorphism that was not possible with more limited genomic information. The sequencing of the genomes of more dioecious plants may define the diversity of dimorphisms that have evolved into the many separate linages of dioecious members. The genomes of other plants with unusual reproductive biology, such as apomictic plants, have not yet been studied but may explain the adaptive value of these modes of reproduction. An analysis of transcriptomes defined the conserved genes for organogenesis that are associated with reproduction in flowering plants [94].

### 5.4. Evolutionary Insights

The comparison of the genome sequences of plant species is the key basis that we have for defining evolutionary relationships. The phylogenetic comparison of plant genomes has often been based on the analysis of one or a few genes. The availability of whole genome sequence data has allowed the use of much larger numbers of genes. A common approach has been to compare the sequences of many conserved single-copy genes. This approach has been used to define relationships between species in the rice (*Oryza*) [86,95] and sorghum (*Sorghum*) [85] genera.

Genome sequencing provides an opportunity to better understand the process of domestication, through which human selection has resulted in plants that are better suited to survival in agricultural environments rather than though natural selection in nature. The sequencing of the coffee genome revealed that human attraction to caffeine might have resulted in the domestication of plants for coffee, tea and chocolate production from diverse species in which caffeine had evolved separately [93].

### 5.5. Maternal Genome Inheritance

In most plants, the cytoplasmic organelles (chloroplasts and mitochondria) are maternally inherited. In some rare cases, other patterns of inheritance are observed; for example, in cucumber (*Cucumis sativus*), the chloroplasts are inherited maternally while the mitochondria are paternally inherited [96]. Although some plant groups display paternal inheritance, the maternal inheritance of organellar DNA and the paternal contribution to nuclear DNA can result in discordant patterns of evolution. Reticulate evolution, in which there is a gene flow between different linages, may result in the transfer (or "capture") of organellar genomes from closely related species. Many examples have been reported via chloroplast genome analyses that suggest phylogenies that differ from those indicated by the nuclear genomes [87]. The potential for the nuclear, chloroplast and mitochondrial genomes of plants to have evolved along separate paths makes it import to focus phylogenetic analysis on nuclear genomes. Improved methods for applying phylogenetic analysis to whole nuclear genomes are needed. Because nuclear genomes show so many variations at the sequence level, the current approaches rely on aligning a small subset of highly conserved genes from nuclear genomes that can be reliably aligned. As whole genome sequence data become more widely available, it may be possible to develop tools that take advantage of the larger volumes of data to better determine relationships through phylogenetic analysis.

*5.6. Importance of Genome Size*

Plant genomes vary widely in size, with more than a 1000-fold variation among flowering plants. Even within a single plant family, genome sizes may vary more than 100-fold. The biological significance of these variations remains poorly understood. Differences may be due to gene content, genome duplications, polyploidy or differences in repetitive DNA content. Improvements in genome sequencing technology are likely to allow larger numbers of diverse genomes to be sequenced to facilitate our understanding of genome size diversity.

**6. Enabling Plant Breeding**

The availability of plant genomes facilitates the breeding and selection of plants, which is essential to support ongoing food security. The need for an accelerated genetic improvement of plants has been made more urgent by climate change, which is demanding new plant genotypes that are adapted to new and more difficult environments. Climate change is altering the physical environment, with higher temperatures and greater water stress, and it is also changing the biological environment, with a wider range of pests and diseases [97].

Plant genomics is critical for the management of plant genetic resources in seed banks [18] and the conservation of the wild crop relatives that provide the genetic resources that are required for sustainable food production.

Plant genomes also enable an understanding of plant biology and the molecular and genetic basis of plant traits.

The analysis of plant genome sequences has facilitated the rapid identification of the key genes that determine the traits of great importance in food crops. This has allowed for more efficient crop breeding by simplifying the selection of these critical traits. The completion of the rice genome allowed for the discovery of the identity of the recessive gene for fragrance in rice [98], which is a trait that can double the value of the rice and can now be detected with a perfect marker [99]. The cooking temperature (gelatinization temperature) of rice was also shown to be determined by a major gene [100]. In wheat, transcriptome analysis has revealed the genetic basis of flour yield [101] and loaf volume for bread [102].

*6.1. Molecular Markers and Plant Selection*

Sequencing has changed the approaches to marker development and applications in plant breeding. Traditional molecular markers [103], which were linked to traits by being close to the genes that control the phenotype, are now able to be replaced by sequence-based markers [104] for the differences in genomes that are actually responsible for the traits under selection. This greatly increases the reliability of the selection. As the costs for whole genome sequencing reduce, this becomes an alternative to the analysis of large numbers of markers. In a sense, genome sequences are the ultimate genetic marker tool as they capture all of the variations within the genome that may explain the phenotype. They also avoid decisions on which genetic markers to select for analysis needing to be made in advance by capturing all of the possibilities.

*6.2. Genetic Manipulation*

Genome sequencing is the ultimate method for the characterization of genetically modified plants, revealing the exact changes in genomes that have been produced. This may be a requirement for release into the environment in some jurisdictions. The sequencing of transgenic genotypes defines both the exact point of insertion and the sequence of the added genes, but it also defines the number of copies that have been added and may reveal any other genetic changes that might have been a result of cell culture [105].

### 6.3. Editing Plant Genomes

The growing application of genome editing is being aided by the use of genome sequencing [106] to support the better targeting of gene editing and to ensure the avoidance of off-target effects. The routine sequencing of the genomes of transgenic- or gene-edited plants can also be used to confirm that the intended changes have been made and that no other unintended changes have occurred. Often, only a single nucleotide needs to be edited or many different loci need to be targeted simultaneously to make the required change in a phenotype; however, in each case, it may be necessary to sequence the genome to confirm that other changes have not occurred.

### 6.4. Biotechnology Applications (Food, Medicinal and Industrial Crops)

Genome sequencing is a key tool that enables the rapid production of plant varieties with higher nutritional value, enhanced levels of bioactivity, improved biomass composition [107] or the expression of high-value molecules. The availability of sequencing technology allows for the identification of novel genotypes with the desired traits and supports the manipulation of plant genomes to produce plants with novel traits. Genome sequencing can be used to confirm the results of gene editing or any other changes resulting from plant transformation, mutagenesis or conventional crossbreeding. Sequencing may be critical for ensuring compliance with regulations that govern plant genetic manipulation. This may become a major application for plant genome sequencing as gene editing becomes more routine. This application for plant genome sequencing may require the development of standard protocols that can be applied routinely, both in research and in industry, for quality assurance and to protect intellectual property.

### 7. IP Issues

An issue of growing importance is the ownership and control of genome sequence data [108]. Modern biological science has been built upon the widespread dissemination of sequence data by providing public access to large sequence databases. The Convention on Biological Diversity empowered countries to claim ownership of their biological resources and the more recent Nagoya protocol requires the consideration of access and benefit sharing. These rights may extend to DNA sequence data that were derived from genetic resources. These issues are especially difficult for historical collections because prior informed consent cannot be obtained for the sample collection [109]. International efforts to resolve these issues are urgently required to balance the rights of traditional owners with the need for open access to sequence data to advance biological science. The protection of plant varieties through the use of plant breeders' rights (PBRs) may be supported by the use of genome sequence data to confirm the identity of genotypes. Establishing the distinctness of genotypes to secure PBRs may also become more dependent on genome sequence data. Genome sequence data may be critical for determining whether a new variety was essentially derived from an earlier variety, which is a question that becomes more important as genome editing becomes more widely used.

### 8. Future Prospects

Many visionary projects aim to ultimately sequence all of the planet's biodiversity as a long-term goal. The sequencing of rare and endangered species can be considered as a priority in this process. Advances in technology have made it difficult to determine when to start such a project as the costs continue to drop and the quality of the data continues to improve. The achievement of plant genome sequences for all species on the planet could accelerate as sequencing technology advances and data storage and handing become more effective.

The difficulty in obtaining a suitable sample of DNA from a plant is one of the remaining challenges in the widespread application of the sequencing of plant genomes. The development of more general methods that can be applied to a wide range of plant

samples would represent a major advance, unless DNA sequencing methods become more robust and can cope with poorer quality DNA preparations.

The technology that is now available for plant genome sequencing and assembly make this an increasingly cost-effective strategy for improving our understanding of the biology of all plant species and a key tool for the conservation of plant biodiversity and the use of plants in agriculture and food production. The sequencing of all plant species is a long-term goal that may become key to effectively supporting life on Earth through the improved management of plants in wild populations and their selection and genetic enhancement for use in agriculture and food production. Threats to food security from human conflicts and pandemics [110] have created more interest in food supply from local sources. Plant genome sequencing provides a platform for innovation in plant breeding to deliver a diverse and balanced diet regionally. Adaptations to climate change require the development of plant varieties that can support the adaptation and relocation of agriculture and the development of plant varieties for production in vertical farming [111]. Genomics is a key tool for tackling climate change [112] and for capturing a wider range of diversity from wild crop relatives [113] and other plants to support food production [114].

**Funding:** This research received no external funding.

**Institutional Review Board Statement:** Not applicable.

**Informed Consent Statement:** Not applicable.

**Data Availability Statement:** Not applicable.

**Acknowledgments:** The author acknowledges the support of the ARC Centre of Excellence for Plant Success in Nature and Agriculture.

**Conflicts of Interest:** The author declares no conflict of interest.

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
