# Peer review of "Progress in Plant Genome Sequencing"

_2813-0464, doi:10.3390/applbiosci1020008_

Round 1

Reviewer 1 Report

This well-written article is a concise overview of the state of the art in plant genome sequencing, containing all the basic information in the subject. There are lot of studies in this area of research, so the references indicated by the Author in the paper are a fairly random subjective selection from this variety. It is desirable in these cases to provide more references to review articles.

Since only data on angiosperms are considered in this work, the title of the article should be modified as: “Progress in Flowering Plants (or Angiospderms) Genome Sequencing”

It would be desirable to add a section on single-cell genomics and transcriptomics.

The Introduction reports that the 2018 Review lists 236 sequenced angiosperm genomes, and Line 190 reports that >100 of them at the chromosome level. Would like to see data that are more recent. To date, the NCBI database contains the genomes of 831 species of Magnoliopsida, of which 373 species have assembly at the chromosome level.

On Lines 96-97 the Author wrote that “Current technologies require a significant amount of DNA”. It is difficult to agree with this; for example, in the cited paper [16], the amount of DNA when comparing different sequencing methods was from 0.01 to 15 μg.

Author Response

Responses: 

More references have been added

Most genomics of plants is focused on seed plants and specifically flowering plants but details of gymnosperm examples have now been included.

Single cell genomics and transcriptomics (now mentioned) has not been applied to plants to any significant extent largely due to the difficulty of single cell isolation from plants without significant changes in expression.

Updated information from NCBI is now included.

The quantity of DNA required for long read sequencing has been greater than that for short reads. More details and a reference is now included.

Reviewer 2 Report

In this review, author comprehensively reviewed the recent Progress in Plant Genome Sequencing technologies. Overall, the review is of great interest and will be useful for early researchers to know insights into sequencing tools in plants. The text also reads very well. However, I noticed that many paragraphs need to be supported by suitable references. I have highlighted some of the places. I hope the author can go through the whole text and add references for vital statements.

Line 44-45, please provide the flowering plant's name, and their genome ranges from 100kb to >100Gb.

Sections 3.3., 4.6., 4.6.2., 4.7., etc., should have some suitable references.

Line 312, which methods? Please provide some examples.

A comprehensive and insightful figure would be a nice addition to give an overview of sequencing technologies. 

Author Response

Response:

Line 44-45 A reference to the range of genome sizes has now been inserted with many species in these size ranges.

Sections  etc  More references have now been added to these sections 

Line 312 A reference has now been provided. The type of methods is now mentioned. The methods are detailed in the following sections.

A figure has been added as suggested.

Reviewer 3 Report

This review paper by R. J. Henry describes the advances in plant genomics, the outcomes and how this new knowledge could help the progress of plant science. In a few pages, the author manages to successfully present the advancement of sequencing technologies, the ways in which plant genomic information is used in plant science and how this aides understanding of species’ biology. Ways in which plant breeding is benefited from this progress and future prospects are also stated. The review is well written and structured, most plant genomics aspects are covered sufficiently and there is adequate referencing that does not make the review long. There are no drawbacks; the only minor point is that Tables should be presented/styled as tables (rather than tab-delimited text). Some minor and sporadic grammar/spelling errors should also be corrected.

Author Response

Response: 

The Tables can be formatted as required by the journal

The manuscript has been checked and grammar/ spelling errors corrected.

Round 2

Reviewer 2 Report

The authors have addressed all the comments, and now the paper can be accepted for publication.